# Body Composition Assessment in Mexican Children and Adolescents. Part 1: Comparisons between Skinfold-Thickness, Dual X-ray Absorptiometry, Air-Displacement Plethysmography, Deuterium Oxide Dilution, and Magnetic Resonance Imaging with the 4-C Model

**DOI:** 10.3390/nu14051073

**Published:** 2022-03-03

**Authors:** Desiree Lopez-Gonzalez, Jonathan C. K. Wells, Alicia Parra-Carriedo, Gladys Bilbao, Martín Mendez, Patricia Clark

**Affiliations:** 1Clinical Epidemiology Research Unit, Hospital Infantil de México Federico Gomez, Mexico City 06720, Mexico; 2Faculty of Medicine, Universidad Nacional Autonoma de México, Mexico City 04510, Mexico; 3Childhood Nutrition Research Centre Population, Policy and Practice, UCL Great Ormond Street Institute of Child Health, London WC1N EH, UK; jonathan.wells@ucl.ac.uk; 4Health Department, Universidad Iberoamericana, Mexico City 01219, Mexico; alicia.parra@ibero.mx (A.P.-C.); gladys.bilbao@ibero.mx (G.B.); 5Centro Nacional de Investigacion en Imagenologia e Instrumentacion Medica, Mexico City 09340, Mexico; mramirez@ci3m.mx

**Keywords:** body composition, dual X-ray absorptiometry (DXA), skinfold-thickness, deuterium dilution, air-displacement plethysmography (ADP), MRI 6, children and adolescents

## Abstract

The evaluation of body composition (BC) is relevant in the evaluation of children’s health-disease states. Different methods and devices are used to estimate BC. The availability of methods and the clinical condition of the patient usually defines the ideal approach to be used. In this cross-sectional study, we evaluate the accuracy of different methods to estimate BC in Mexican children and adolescents, using the 4-C model as the reference. In a sample of 288 Mexican children and adolescents, 4-C body composition assessment, skinfold-thickness (SF), dual-energy X-ray absorptiometry (DXA), air displacement plethysmography (ADP), and deuterium dilution (D_2_O) were performed, along with MRI in a subsample (52 participants). The analysis of validity was performed by correlation analysis, linear regression, and the Bland–Altman method. All methods analyzed showed strong correlations for FM with 4-C values and between each other; however, DXA and MRI overestimated FM, whereas skinfolds and ADP under-estimated FM. Conclusion: The clinical assessment of BC by means of SF, ADP, DXA, MRI and D_2_O correlated well with the 4-C model and between them, providing evidence of their clinical validity and utility. The results from different methods are not interchangeable. Preference between methods may depend on their availability and the specific clinical setting.

## 1. Introduction

In Mexico, 35.6% and 38.4% of children and adolescents have overweight or obesity (OW/OB) according to the 2018 Mexican National Health and Nutrition Survey (ENSANUT) [1,2]. OW/OB represent a key public health problem in the country and are closely related to the top three causes of mortality (heart diseases, diabetes, and cancer) [3].

Although OW/OB are defined as the excessive accumulation of adipose tissue leading to increased risk of negative health outcomes, they are routinely categorized in all age groups using body mass index (BMI) [4,5].

BMI is almost universally adopted to assess nutritional status because of its simplicity, practicality, and its good correlation with adipose tissue [6,7]. Nevertheless, the limitations of using BMI for this purpose are increasingly recognized [8,9,10], and it should not be used as a standard in the specific assessment of adiposity [11]. BMI may be imprecise and may misclassify those of short and tall stature and those with a significant increment of their muscle mass [12]. BMI may also be insensitive to change by interventions (e.g., reductions in adipose tissue coupled with increments in muscle mass related to successful nutritional and physical activity intervention may not equate to changes in BMI) [13]. Finally, individuals with high levels of lean mass (constituted mostly of functional and useful tissues that are not necessarily unhealthy) may be misclassified as OW/OB due to high BMI [14,15].

In addition to OW/OB, several other health conditions (e.g., cancer, malnutrition, storage diseases, chronic exposure to systemic corticosteroids, etc.) are also associated with alterations in weight which may be insufficiently described by BMI [16,17,18].

The increased availability of alternative technologies has brought significant improvements in our capacity to measure human-body physical characteristics. Likewise, the assessment of the relative distribution of different tissues that contribute to body composition (BC) has gained interest and relevance in the evaluation of health and disease [19].

Clinical BC acknowledges four different components of weight: fat mass (FM), protein mass (PM), bone mineral content (BMC), and total body water (TBW). Currently, the estimation of these components is possible using a multi-technique approach known as the 4-component (4-C) model. While this is the most accurate method, its complexity, cost, time to provide results, and exposure to radiation challenges its use for routine clinical practice [20,21]. For these reasons, the 4-C model is only used for research purposes. Simpler, faster, safer, and less expensive techniques are more readily available and are increasingly used in the clinical assessment of BC. Each technique has its advantages and its limitations and thus may be applicable to different scenarios.

A key issue concerns variability in the accuracy of the different techniques. The aim of this study was, therefore, to compare several BC estimation methods to the criterion 4-C model in Mexican children and adolescents to better inform the clinicians about their performance, ultimately facilitating the adoption of the most suitable method to assess BC.

## 2. Materials and Methods

### 2.1. Participants

Healthy children and adolescents that participated in the study “Reference values of body composition of Mexican children and adolescents” were invited to participate in this study using an age- and sex-stratified random procedure. Our approach was described in detail previously [22]. Briefly, it was a population-based cross-sectional study of more than 1500 volunteer healthy Mexican children and adolescents who were residents of Mexico City. These participants were clinically, nutritionally, and biochemically assessed to confirm their health status prior to the corresponding measurements, with the objective to describe the reference values of body composition for Mexican children and adolescents [22]. Sampling selection for the current study was performed using this database with an iterative stratified random process considering age in yearly intervals from 4 to 18 years and sex. A sample size of 7 participants per year of age and sex was calculated as appropriate for an expected correlation coefficient of ≥0.90 with a two-tailed type 1 error rate of 5% and type 2 error rate of 20% [23]. Recruitment was conducted by telephone calls, where the study was explained in detail and carried out from June 2018 until the sample size was completed for each age and sex group, which occurred in July 2019. This study was reviewed and approved by our institutional ethics, biosafety, and research committees (Registered as HIM 2015-055).

### 2.2. Measurements

#### 2.2.1. Clinical Assessment

Invited children and adolescents that agreed to participate were asked to arrive at the study site after 8 h fasting for the measurements. All parents or guardians of participants signed an informed consent form, and children aged ≥ 7 years were asked for their assent as well. Participants were clinically, nutritionally, and biochemically assessed by a pediatrician and a nutritionist to confirm their health status. Pubertal development stage was registered according to the Tanner and Whitehouse scale [24,25]. Those with biochemical abnormalities were not included in this study (i.e., impaired fasting glucose; low high-density cholesterol; high triglycerides; or insulin resistance according to the Expert Panel on Integrated Guidelines for Cardiovascular Health and Risk Reduction in Children and Adolescents criteria) [26].

#### 2.2.2. Anthropometry

Weight and height were measured with participants wearing lightweight clothing, using a SECA^®^ 284 scale stadimeter. Waist and hip circumferences were measured according to WHO standards using a SECA^®^ 201 measuring tape [27]. Mid upper arm, thigh, and calf circumferences were measured according to the International Standards for Anthropometric Assessment by the International Society for the Advancement of Kinanthropometry (ISAK) recommendations [28].

BMI was calculated as weight (kg) divided by the square of height (m) [29]. Weight, height, and BMI z-scores were calculated using the growth reference of the World Health Organization [30].

#### 2.2.3. Skinfold Thickness

SF thicknesses were measured according to Lohman’s technique following ISAK recommendations [28]. They were measured at the triceps and calf, twice for each site and for both body sides, with a calliper with a scale of 0–80 mm and precision of ±0.2 mm (Harpenden calliper British Indicators Ltd., St Albans, UK). Measurements were taken to the nearest millimetre at each site, and the mean of the four values for each region was calculated. The percentage of fat was calculated according to the equations of Slaughter et al. and multiplied by the total weight of each subject to obtain total fat mass [31]:Males Percentage of fat (%) = 0.735 (triceps + calf) + 1.0
Females Percentage of fat (%) = 0.610 (triceps + calf) + 5.1
Total Fat-mass = fraction of fat × weight (kg)

#### 2.2.4. Dual X-ray Absorptiometry (DXA)

A whole-body scan was performed on all participants using a Lunar-iDXA densitometer (GE Healthcare^®^) according to the manufacturer’s instructions and analyzed through ENCORE^®^ software version 15. Measurements were performed by an International Society of Clinical Densitometry (ISCD)-certified nurse, and calibration of the densitometer was performed on a weekly basis according to the manufacturer’s instructions. DXA total body composition assessment with regional analysis provided data for total body (with head) fat mass (FM), lean soft tissue mass (LM) and bone mineral content (BMC) [32], and regional from arms, legs, and trunk [33]. DXA FFM values were calculated as total body LM plus BMC.

#### 2.2.5. Air-Displacement Plethysmography (ADP)

Body volume was measured by ADP using BOD POD^®^ instrumentation (COSMED USA Inc., Concord, CA, USA, Software version 5.2.3) with standardized procedures according to the manufacturer’s instructions [34]. Briefly, participants had to abstain from physical activity and food 2 h before the measurement. The BOD POD was calibrated each day before use according to the manufacturer’s guidelines. Study participants were measured in tight-fitting bathing suits with swimming caps to minimize air trapped in clothing and hair. Body mass was measured using the BOD POD’s precise electronic scale, while body volume was measured in the chamber twice. If the first two readings for body volume differed by more than 150 mL, a third measurement was taken, and the two values that were closest and within the criteria for the agreement were averaged. Thoracic gas volume (TGV) was predicted by the software with a validated child-specific equation [34,35]. The fat–mass percentage (FMADP%) and fat mass by ADP (FMADP) were calculated using up-to-date child-specific conversion factors reported included in the paediatric software [35,36].

#### 2.2.6. Deuterium Oxide Dilution (D_2_O)

Following ≥8 h fasting, a pre-dose saliva sample was collected, then a dose of 0.05 g per kg weight of D_2_O diluted in 50 mL of tap water was given to drink, and 4 h after, a post-dose saliva sample was taken. TBW was calculated from D_2_O dilution according to Buchholz et al. with the following formula [37]:$$\mathrm{TBW}\;\mathrm{in}\;\mathrm{kg}\;=\frac{\frac{D_{2}O\;\mathrm{dose}\cdot \mathrm{concentration}}{20}\cdot \frac{18.02}{{\mathrm{AP}}_{\mathrm{post}-\mathrm{dose}}-{\mathrm{AP}}_{\mathrm{pre}-\mathrm{dose}}}\cdot 10^{-3}}{1.04}$$
$$\mathrm{TBW}\;\mathrm{in}\;\mathrm{liters}\;=\frac{\mathrm{TBW}\;\mathrm{in}\;\mathrm{kg}}{0.99371}$$
TBW (kg or L): total body waterD_2_O dose (g): deuterium (D_2_O)Concentration (%): atom percent of supplied D_2_O: 99.9%20: molecular weight of D_2_O18.02: molecular weight of tap waterAP_post-dose_ (%): atom percent of D_2_O in post-dose saliva sampleAP_pre-dose_ (%): atom percent of D_2_O in pre-dose saliva sample10^−3^ (kg/g): calculation from g to kg1.04: correction for proton exchange0.99371 (kg/L): density of water at 36 °C
where the D_2_O dose was calculated as follows:$$D_{2}O\;\mathrm{dose}=\left({\mathrm{bottle}}_{D_{2}O}-{\mathrm{bottle}}_{\mathrm{empty}}\right)\cdot \frac{{\mathrm{bottle}}_{D_{2}O\;\mathrm{and}\;\mathrm{water}}-{\mathrm{bottle}}_{\mathrm{drunk}}}{{\mathrm{bottle}}_{D_{2}O\;\mathrm{and}\;\mathrm{water}}-{\mathrm{bottle}}_{\mathrm{empty}}}$$

bottle _empty_—an empty bottle was weighed with an Ohaus scale model PA4202C with 0.01 g precision and accuracy of two decimal places.bottle _D_2_O_—0.05 g of D_2_O per kilogram of body weight were filled into the bottle, which was weighed again.bottle _D_2_O and water_—50 mL of tap water were added, and again the bottle was weighed.bottle _drunk_—participants drank the D_2_O with tap water, and the bottle was weighed again.

D_2_O doses were outliers and not used for calculations of TBW when the weight of the bottle was lower after drinking the D_2_O with tap water than it was in the beginning, when the remaining D_2_O with tap water after drinking was more than 1 g, or when the weighed D_2_O differed more than 15% from the target dose of 0.05 g of D_2_O per kilogram of body weight.

#### 2.2.7. Magnetic Resonance Imaging (MRI)

As an exploratory part of this study, we measured total-body fat mass by whole-body multi-slice MRI for a subsample of participants from this study (*n* = 52). Participants were placed in a 3.0 Tesla (T) scanner (Achieva 3.0T, Philips Medical Systems, Best, The Netherlands) in a supine position with their arms by their sides. T1-weighted (TR/TE: 72.3/2.3 ms) and T2-weighted (TR/TE: 1093.4/76 ms) coronal images (6 mm slice thickness, 1.0 mm gap) were acquired across the whole body. The intervertebral space between the fourth and fifth lumbar vertebrae (L4–L5) was set as the point of origin for abdominal T2-weighted (TR/TE: 3000/16 ms) water suppression and transverse images (8 mm slice thickness, 1.0 mm gap) covering the abdominal area. We then calculated the size of voxels, counted those with fat, and multiplied them by the adipose tissue density to obtain a value for total body fat mass. Visualization, annotation, and quantification were performed in MATLAB R2020b (The Mathworks, Inc., Natick, MA, USA).

#### 2.2.8. The 4-Compartment Model (4-C)

The 4-C model was used as the reference standard method for the estimation of fat mass and was calculated according to Fuller et al. [38]:$$\mathrm{FM}=2.7474\mathrm{BV}-0.7145\mathrm{TBW}+\frac{1.4599\mathrm{BMC}}{1000}-2.0503\mathrm{weight}$$

FM (kg): fat massBV (L): body volume measured by ADPTBW (L): total body water measured by D_2_O dilutionDMC (g): bone mineral content measured by DXAWeight (kg): body weight


FFM=weight−FM


FFM (kg): fat-free massWeight (kg): body weightFM (kg): fat mass

### 2.3. Statistical Analysis

Descriptive statistics were used to characterize the demographics and measurements of each method, expressing results as means and standard deviations for continuous variables and percentages for categorical variables.

The means of FM estimated by each method in comparison with the 4-C model, for the total sample and by age and sex groups, were compared using a paired *t*-test.

Pearson correlation coefficients and Lin’s concordance correlation coefficients were computed for the estimated FM, %FM, and FFM values by each method with respect to the reference standard of the 4-C model. A simple linear regression was performed to determine the relationship between body composition methods and obtain the equation for each method with the 4-C model for total body FM and FFM estimation. The Bland–Altman method [39] was used to assess agreement between each method with the 4-C model as reference standard. In this procedure the differences of FM values estimated by each method minus the values estimated by the 4-C model (y-axis) were plotted against the average of such two measurements (x-axis). The means of FM estimated by the different methods were compared by paired *t*-tests. The mean difference and the limits of agreement (+/−2 SD of the difference) were calculated, and linear regression analysis with the difference as the dependent variable, and the average of measurements as the independent variable, were undertaken for each method to assess proportional bias (i.e., whether the magnitude of the bias varied depending on the level of fatness) [39].

Statistical analyses were conducted in SPSS for Windows version 21.0 (SPSS Inc., Chicago, IL, USA) and Prism 8 for Windows (GraphPad Software, Inc., San Diego, CA, USA). Statistical significance was set at *p* < 0.05.

## 3. Results

A total of 293 children and adolescents were measured; data from 5 participants met the outlier criteria for TBW measurement by D_2_O and were not included. We report results from the measurements of 288 participants (aged 4 to 18 years old); 53% of them were females, and 173 (59%) were adolescents (11 to 18 years). Demographics and measurements data are summarized in Table 1. For clarity, all data are presented by sex and age group unless otherwise specified. The characteristics of the subjects measured with MRI are shown in Appendix A.

Comparisons of mean FM values estimated by each method in comparison with the 4-C model for the total sample and stratified by age and sex group are shown in Appendix A. ADP FM mean values for the whole sample were consistently and significantly lower than those estimated by the 4-C model (8.2 ± 6.5 kg vs. 9.5 ± 6.8 kg), whereas those estimated by DXA and MRI were consistently and significantly higher (12.5 ± 6.8 kg and 12.9 ± 5.7 kg, respectively). FM estimated by D_2_O was similar to 4-C values in children but significantly higher in adolescents. The estimation of FM by SF showed mean values that were significantly lower for female children and adolescents, significantly greater for male adolescents, and similar in male children when compared to the 4-C mean values.

The correlations, concordances, agreements, and proportional bias assessments of FM between SF, DXA, ADP, D_2_O, and MRI with respect to the 4-C model are shown in Table 2.

All methods showed strong to very strong correlations with 4-C values for FM (i.e., Pearson’s correlation coefficients > 0.80) across all age and sex groups. For this study, we used the Slaughter formula to estimate BC from SF measurements, but in Appendix A we also provide raw data and correlation analyses for raw SF data as additional analyses.

Lin’s concordance correlation coefficients ranged from poor (<0.90) to substantial (>0.95) precision and accuracy for each method in comparison with the 4-C model across age and sex groups. The Bland–Altman agreement analyses disaggregated by sex and age groups showed the lowest mean bias for D_2_O (−0.17 to +0.94 kg) and SF (−1.2 to +0.74 kg) with respect to the 4-C model, and greater values for MRI (1.37 to 3.1 kg), ADP (−2.1 to −0.57 kg) and DXA (2.58 to 3.2 kg).

When the full range of measurements (both sexes and all age groups) were analyzed in the Bland–Altman plots, again SF and D_2_O showed the least bias with −0.34 kg (LOA −5.2, 4.5 kg) and 0.52 kg (LOA −4.3, 5.4 kg), respectively. Results of SF, ADP, and MRI showed significant negative trends for the bias, with increasing FM overestimation in those with higher FM. Conversely, D_2_O showed the opposite trend with increasing FM underestimation in those with higher FM. Greater bias values (2.95 kg; LOA −1.1, +7.0 kg) were evident for DXA, MRI (2.28 kg; LOA −3.2, +7.8 kg) and ADP (−1.35 kg; LOA −5.3, +2.6 kg). Only DXA showed consistent behaviour across all measured values, as shown in Figure 1.

In the FFM estimation for the total sample, all methods showed good correlations with the 4-C model (Pearson ≥ 0.97), and Lin’s concordance correlation coefficients (≥0.95) indicated high precision and accuracy. In the Bland–Altman plots, SF and D_2_O showed the least bias with 0.31 kg (LOA −4.5, +5.1 kg) and −0.53 kg (LOA −5.4, 4.3 kg), respectively. Results of SF and DXA showed significant positive trends of the differences, with increasing FFM underestimation in those with higher values. In contrast, ADP, D_2_O, and MRI showed consistent behaviour across all measured values. The correlations by age group and sex are shown in Table 3.

Lin’s concordance correlation coefficients for precision and accuracy ranged from poor (<0.90) for DXA to substantial for D_2_O (~0.95) in comparison with FFM by the 4-C model across age and sex groups. The Bland–Altman agreement analyses disaggregated by sex and age groups showed the least mean bias for D_2_O (−0.86 to +0.21 kg) and SF (−0.74 to 1.0 kg) with respect to the FFM with the 4-C model, and greater values for DXA (−3.3 to −2.7 kg) and ADP (0.42 to 1.28 kg). The Bland–Altman plots for these analyses for the total sample are shown in Figure 2.

Further analyses of correlation, concordance, and agreement between the different methods in the estimation of FM are shown in Appendix A.

In the correlation analysis, all the different techniques showed r values ≥ 0.83. The techniques with the best concordance were DXA with MRI and SF with ADP. The main differences were between MRI and D_2_O. In the Bland–Altman analyses, all three techniques showed significant biases in the mean estimation of FM, as shown in Appendix A.

## 4. Discussion

This study compared five different BC estimation methods to the 4-C model in Mexican children and adolescents. For this study, the 4-C model was considered the reference standard of BC assessment. Like previous publications, our results showed that all five methods provided data on FM and FFM that correlated well with the 4-C model [36,40,41,42]. However, Lin’s concordance correlation coefficients and Bland–Altman plots provided more detailed information regarding significant differences between methods. According to such analyses, D_2_O, SF, MRI, and ADP showed the highest overall concordance and the lowest bias, though with higher FM values, proportional biases became significant and agreement between each of these methods and the 4-C model decreased. In contrast, DXA consistently overestimated FM by approximately 3 kg, but this was stable across the different values of FM, showing lower accuracy but higher precision than D_2_O, SF, and ADP.

Considering their availability, accessibility, and affordability, measuring SF may represent a preferred choice for clinicians across different levels of healthcare. This method requires the least infrastructure investment, is non-invasive, reproducible, relatively comfortable for the patient, may be repeated as frequently as required without risk, and can be conducted in the clinical setting of critically ill patients and those with mobility restrictions [43]. Limitations include the need for trained and standardized personnel, the method is operator-dependent, entails the application of an equation, or conversion to Z-scores, and its performance may be compromised in clinical conditions where BC assessment is frequently required, such as oedema, extreme obesity, and other conditions such as muscular or lipid dystrophies, storage diseases, among others [44]. In our data, SF showed variations in the estimation of FM in the range of limits of agreement from −5.2 to +4.5 kg in comparison to the 4-C model. SF showed a significant proportional bias with increasing sub-estimation for increasing FM values (beta-coefficient −0.1; *p* < 0.001). The magnitude of the proportional bias may represent a compromise in its clinical performance when assessing patients with the highest FM values (e.g., OW/OB) but may be less relevant for those with malnutrition, cancer, or other conditions, including the nutritional assessment of healthy subjects.

BC by ADP is currently possible only using a single commercially available device known as the BOD POD^®^ (Cosmed USA Inc., Concord, CA, USA). This device has specifically been designed to assess BC and is very popular in weight-management programs and among high-performance athletes. BC assessment by ADP has several advantages: it is non-invasive, relatively easy to perform, reproducible, tightly calibrated, and can also be repeated as frequently as needed without risks. However, it requires a significant investment for the device, a dedicated room with constant temperature and pressure, and trained personnel. ADP is not easily performed in individuals with mobility restrictions or in critically ill patients. Because ADP estimates FM and FFM assuming a constant of tissue hydration, individuals with several diseases affecting hydration status may be inadequately assessed by this method as well. [21]. BC is usually contraindicated in individuals with claustrophobia and may be limited for individuals with excessively large body sizes. The use of skin moisturizers and even abundant hair may compromise its precision and accuracy as well. In our data, ADP showed variations in the estimation of FM in the range of limits of agreement from −5.3 to +2.6 kg in comparison to the 4-C model. ADP showed significant proportional bias with increasing sub-estimation for increasing FM values (beta-coefficient −0.05; *p* = 0.003). Again, the magnitude of the proportional bias may represent a compromise for those with the highest FM values but not so relevant for other health conditions.

DXA has gained substantial interest because of the growing versatility of clinical assessments that can be conducted. Initially, DXA was developed to estimate bone mineral density (BMD), and it is currently the standard clinical tool to diagnose osteopenia or osteoporosis. Subsequently, DXA is increasingly used to estimate other body components, such as FM and lean mass (LM) (i.e., fat-free and bone-free mass) [45]. This method has become popular, and at some centers, it is considered the clinical gold standard for BC assessment [46]. The major advantages of this method are that it allows the estimation of three components (BMC, FM, and LM) in a relatively simple and fast assessment (i.e., <15 min with results immediately available), and it is very reproducible (as long as the same technology is used). It also allows for BC assessment by regions (i.e., arms, legs, trunk), which may be of clinical relevance. The major limitations of DXA include exposure to radiation which impedes repeated frequent assessments, it requires a high investment for the device, related infrastructure, and its maintenance. Individuals with limited mobility, critically ill, those with prosthetics, those who are or might possibly be pregnant, and those unable to stay still during the scans may compromise the feasibility of this type of BC assessment [47,48,49]. In our data DXA showed variations in the estimation of FM in the range of limits of agreement from −1.1 to +7.0 in comparison to the 4-C model. The DXA estimations showed no significant bias across the different values of FM (beta-coefficient −0.006; *p* = 0.74), consistent with other reports in the literature [36,40,49,50].

D_2_O is considered a reference method to estimate total body water. It relies on the ingestion of labelled water, and then by adjusting for hydration coefficients, FM and FFM can be estimated. This method is non-invasive, with no known adverse effects, may be used in pregnant women, children, elderly, and may be used multiple times without clinical consequences [51]. However, this technique requires mass spectrometry analyses or Attenuated Total Reflection Fourier Transformed Infrared Spectroscopy (ATR-FTIR), which necessitates access to such technology and infrastructure, trained personnel, and usually the time from assessment to results may be considerable. In our data, D_2_O showed variations in the estimation of FM in the range of limits of agreement from −4.3 to +5.4 in comparison to the 4-C model. D_2_O showed a significant proportional bias with increasing supra-estimation for increasing FM values (beta-coefficient +0.09; *p* < 0.001).

MRI offers an interesting approach to BC given its ability to discriminate between different tissues, offering a unique perspective on body fat mass [52]. This method is non-invasive and may be repeated in the follow up of patients. MRI also allows for the assessment of adipose tissue in specific regions and organs. However, MRI may represent a challenging tool for BC assessment in the clinical setting. It requires a significant investment of the device-related infrastructure and maintenance; it usually necessitates a considerable amount of time for image acquisition, where participant cooperation is needed, and as with ADP, claustrophobia may be a relative contraindication. Individuals’ size, mobility restrictions, prosthetics, and critical illness may also prevent this method from being used. Moreover, MRI interpretation may require considerable time trained personnel and pose challenges to the time taken from images acquisition to a clinical result. For such reasons, and even in the research setting such as this study, BC assessment by MRI presents major challenges. In our data, MRI showed variations in the estimation of FM in the range of limits of agreement from −3.2 to +7.8 in comparison to the 4-C model. MRI also showed a significant proportional bias with increasing sub-estimation for increasing FM values (beta-coefficient −0.1; *p* = 0.03).

Our results provide relevant data to clinicians regarding the acceptable clinical performance of all analysed techniques compared to the 4-C model. In addition, this study also compared correlations, concordances, and agreements between such different methods. Routine clinical practice may assess BC by means of SF, DXA, ADP or MRI, whereas D_2_O and the 4-C model are mostly conducted for research purposes. As this study has shown, the methods have good correlations with the 4-C model and between each other; but their results are not interchangeable therefore clinical assessments of BC, especially where follow-up is relevant, should be made with the same technique to avoid unproper comparisons.

Limitations of the current study include the sample being restricted to healthy participants from 4.5 to 18 years of age. In this study, we only present data from Hispanic subjects living in Mexico City and its Metropolitan Area. Those from rural areas, Afro-Mexican, and indigenous populations, as well as from other territories of Mexico, may not share the same characteristics of our sample. Therefore, we advise caution when comparing subjects from such groups. As the Bland–Altman plots showed, significant biases were evident, and increasing disagreement was observed at higher values of FM for several methods. This finding may be influenced by the smaller number of participants with very high FM values. Future studies with larger samples of participants with OW/OB may allow for further analyses and ascertain if, in fact, a significant bias is dependent on FM. Another limitation of this study is its inability to capture and compare the clinical performance of assessed methods of BC assessment in specific clinical conditions of interest (i.e., malnutrition, storage diseases and diseases where bone, muscle, adipose tissue and or hydration status are affected may influence on clinical performance of the different methods assessed in this study). Data on MRI was limited because of sample size, so no robust conclusions can be drawn from this study; therefore, data were presented as an exploratory analysis only.

We firmly believe that BC assessment should have a more important role, especially in a population such as ours where OW/OB and other health conditions that impact BC (cancer, malnutrition, chronic diseases, chronic exposure to systemic corticosteroids, etc.) are increasing in prevalence within the paediatric population. Such BC assessment in the clinical setting requires precise, accurate, simple, safe, and accessible methods. As our results showed, none of the methods are ideal compared to the criterion 4-component model, but knowing the magnitude and direction of their biases should aid the clinician in the appropriate tool selection and its potential impact on BC estimation. Practicality, versatility, and time to results are other valuable attributes to consider. This study performed a comprehensive comparative analysis of five different BC assessment methods, providing supportive data for their clinical use, significant differences between them and consistent evidence against their interchangeability. Availability and the specific clinical context might provide further direction on preference between the different methods.

## 5. Conclusions

Clinical assessment of BC by means of SF, ADP, DXA, MRI, and D_2_O correlated well with the 4-C model, providing evidence of the clinical validity and usefulness of these approaches. All of the methods are appropriate for ranking individuals within a population in terms of their FFM and FM, and this information is often of great value in monitoring clinical progress. Significant differences in concordance and agreement were observed between the methods and varied across different values of FM, indicating that the methods cannot be used interchangeably. However, some of the bias associated with any specific technique can be resolved by providing method-specific reference data whereby raw data are converted to z-scores [20], and providing such comprehensive reference data for Mexican children and adolescents is a further aim of this project. Preference between the methods may depend on their availability and the specific clinical setting, but the emphasis should be maintained on the importance of assessing BC in routine care of the pediatric population.

## Figures and Tables

**Figure 1 nutrients-14-01073-f001:**
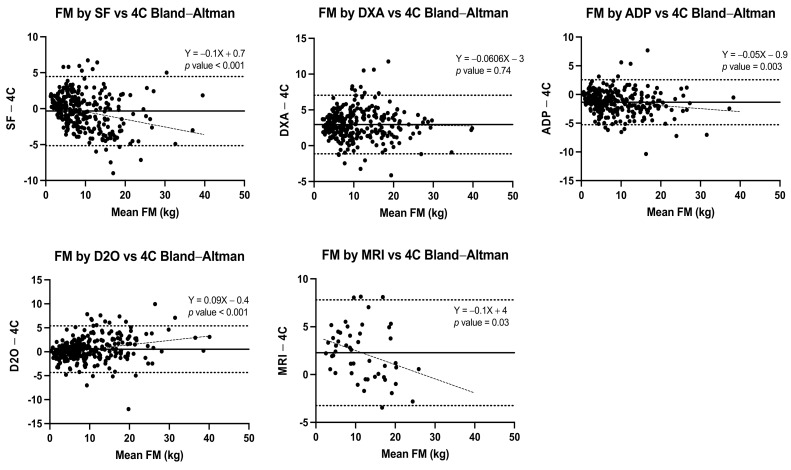
Bland–Altman plots for FM estimation by SF, DXA, ADP, D_2_O, and MRI in contrast with the 4-C model for the total sample. A positive trend indicates an increasing underestimation of FM at high FM levels; a negative trend indicates an increasing overestimation of FM at high FM levels.

**Figure 2 nutrients-14-01073-f002:**
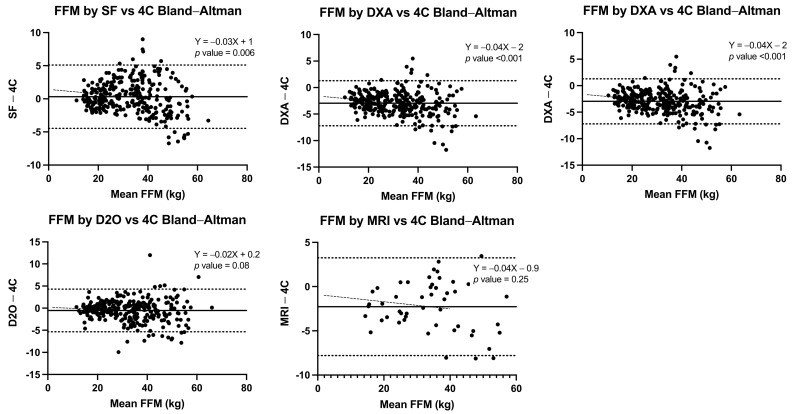
Bland–Altman plots for FFM estimation by SF, DXA, ADP and D_2_O in contrast with the 4-C model for all ages and sex groups. A positive trend indicates an increasing underestimation of FFM at high FFM levels; a negative trend indicates an increasing overestimation of FFM at high FFM levels.

**Table 1 nutrients-14-01073-t001:** Demographic and data on measurements by age and sex groups (*n* = 288).

	Children 4–10 Years		Adolescents 11–18 Years	
	Female *n* = 63	Male *n* = 54	Female *n* = 92	Male *n* = 79
Age (years)	8.1 ± 1.8	8.4 ± 1.7	14.5 ± 2.1	14.8 ± 2.0
Weight (kg)	26.1 ± 8.1	28.6 ± 8.1	51.2 ± 12.0	53.9 ± 11.6
Weight z-score	−0.21 ± 0.97	0.13 ± 1.19	-	-
Height (cm)	123.5 ± 14.7	127.9 ± 11.1	155.1 ± 7.3	163.4 ± 10.3
Height z-score	0.11 ± 1.07	0.53 ± 1.14	−0.44 ± 0.92	−0.25 ± 0.97
BMI (kg/m^2^)	17.1 ± 6.0	17.2 ± 2.6	21.1 ± 3.9	20.0 ± 3.1
BMI z-score	0.11 ± 1.07	0.53 ± 1.14	0.32 ± 1.12	−0.05 ± 1.19
Waist circumference (cm)	57.2 ± 8.0	60.4 ± 7.6	71.3 ± 7.8	72.1 ± 8.8
Tanner				
1	49 (78%)	53 (98%)	1 (1%)	4 (5%)
2	9 (14%)	1 (2%)	4 (4%)	6 (8%)
3	5 (8%)	0	29 (32%)	36 (46%)
4	0	0	45 (49%)	31 (39%)
5	0	0	13 (14%)	2 (3%)
BMI category				
Healthy weight	49 (78%)	40 (74%)	65 (71%)	62 (78%)
Overweight	10 (16%)	9 (17%)	20 (22%)	10 (13%)
Obesity	3 (5%)	5 (9%)	6 (7%)	4 (5%)
Low weight	1 (2%)	0	1 (1%)	3 (4%)
Body composition variables				
SF-sum (mm)	21.2 ± 8.6	22.0 ± 9.4	32.3 ± 12.7	22.1 ± 10.9
SF FM (%)	18.1 ± 5.3	17.1 ± 6.9	24.8 ± 7.7	17.2 ± 8.0
SF-FM (kg)	5.0 ± 3.3	5.5 ± 3.4	13.5 ± 7.2	9.7 ± 6.0
SF-FFM (kg)	21.1 ± 5.2	23.3 ± 5.1	37.9 ± 6.5	44.4 ± 8.9
D_2_O TBW (kg)	15.01 ± 3.7	16.9 ± 3.6	26.6 ± 5.0	31.9 ± 7.2
D_2_O FM (%)	19.7 ± 10.3	18.4 ± 9.7	28.2 ± 10.1	19.3 ± 10.0
D_2_O FM (kg)	5.4 ± 4.5	5.5 ± 4.4	15.2 ± 8.1	10.3 ± 6.4
D_2_O FFM (kg)	20.5 ± 5.1	22.9 ± 4.9	36.1 ± 6.7	43.3 ± 9.8
DXA BMC (kg)	0.9 ± 0.2	1.0 ± 0.2	1.9 ± 0.4	2.1 ± 0.5
DXA FM (%)	29.6 ± 6.2	28.6 ± 6.9	33.6 ± 6.1	23.3 ± 7.9
DXA FM (kg)	8.2 ± 4.2	8.3 ± 4.2	17.5 ± 6.7	12.5 ± 5.8
DXA FFM (kg)	17.8 ± 4.4	20.1 ± 4.3	33.7 ± 5.8	40.8 ± 9.9
DXA LM (kg)	16.9 ± 4.2	19.1 ± 4.1	31.8 ± 5.5	39.2 ± 8.4
ADP BV (L)	24.9 ± 8.3	20.1 ± 9.3	49.1 ± 13.1	50.8 ± 11.3
ADP TGV (L)	1.2 ± 0.4	1.3 ± 0.3	2.3 ± 0.4	2.8 ± 0.6
ADP FM (%)	14.1 ± 8.6	17.0 ± 8.7	22.6 ± 7.9	15.2 ± 8.8
ADP FM (kg)	4.1 ± 4.0	5.2 ± 4.2	12.2 ± 6.9	8.3 ± 5.9
ADP FFM (kg)	23.0 ± 9.8	22.9 ± 5.4	39.2 ± 8.3	45.3 ± 9.1
4-C FM (%)	19.4 ± 8.7	19.1 ± 9.0	26.8 ± 8.4	17.4 ± 8.9
4-C FM (kg)	5.5 ± 4.2	5.8 ± 4.3	14.4 ± 7.3	9.3 ± 5.8
4-C FFM (kg)	20.6 ± 4.7	22.7 ± 4.9	36.8 ± 6.4	44.4 ± 9.8
Hydration (%)	72.9 ± 5.5	73.9 ± 4.4	71.7 ± 5.0	71.6 ± 5.2
MRI subsample (*n* = 52)				
MRI FM (%)	31.0 ± 7.4	25.9 ± 6.2	31.5 ± 5.9	23.6 ± 6.0
MRI FM (kg)	8.8 ± 5.5	7.0 ± 4.2	16.4 ± 4.8	12.6 ± 4.8

BMI = body mass index, SF = skinfold-thickness (triceps + calf), FM = fat mass, FFM = fat free mass, D_2_O = Deuterium oxide dilution, TBW = total body water, DXA = Dual X-ray absorptiometry, BMC = bone mineral content, LM = lean mass, ADP = Air-displacement plethysmography, BV = body volume, TGV = thoracic gas volume, 4-C = four compartment model, MRI = Magnetic Resonance Imaging.

**Table 2 nutrients-14-01073-t002:** Correlations, concordances, agreements, and proportional bias assessments between SF, DXA, ADP, D_2_O, and MRI with respect to the 4-C model for the estimation of FM.

Method	Pearson’s R, (95% CI)	Regression Equation	Lin’s Concordance Coefficient (95% CI)	Bland–Altman Difference Mean ± SD	BA LOA	Regression Analysis from Bland–Altman Differences	*p* Value
Total sample							
SF (*n* = 288)	0.93 (0.92–0.95)	Y = 0.84 (0.80 to 0.88) X + 1.2 (0.72 to 1.6)	0.93 (0.91–0.94)	−0.34 ± 2.5	(−5.2 to 4.5)	Y = −0.1X + 0.7	*p* < 0.001
DXA (*n* = 288)	0.95 (0.94–0.96)	Y = 0.95 (0.91to 0.98)X + 3.4 (3.0 to 3.9)	0.87 (0.85–0.89)	2.95 ± 2.1	(−1.1 to 7.0)	Y = −0.006X + 3	*p* = 0.74
ADP (*n* = 288)	0.96 (0.95–0.97)	Y = 0.91 (0.88 to 0.94)X − 0.48 (−0.86to −0.10)	0.94 (0.92–0.95)	−1.35 ± 2.0	(−5.3 to 2.6)	Y = −0.05X − 0.9	*p* = 0.003
D_2_O (*n* = 288)	0.94 (0.93–0.96)	Y = 1.0 (0.99 to 1.1)X + 0.23 (−0.27 to 0.72)	0.94 (0.92–0.95)	0.52 ± 2.5	(−4.3 to 5.4)	Y = 0.09X − 0.4	*p* < 0.001
MRI (*n* = 49)	0.91 (0.84–0.95)	Y = 0.79 (0.68to0.89)X + 4.5 (3.2to 5.8)	0.4 (0.17–0.59)	2.28 ± 2.8	(−3.2 to 7.8)	Y = −0.1X + 4	*p* = 0.03
Female children							
SF (*n* = 63)	0.93 (0.89–0.96)	Y = −0.22 (−0.92 to 0.48) + 1.6 (1.04 to 1.3)X	0.9 (0.86–0.94)	−0.49 ± 1.6	(−3.6 to 2.6)	Y = −0.2X + 0.7	*p* < 0.001
DXA (*n* = 63)	0.93 (0.89–0.96)	Y = −2.3 (−2.99 to −1.61) + 0.95 (0.88 to 1.03)X	0.79 (0.71–0.85)	2.73 ± 1.3	(0.3 to 5.2)	Y = 0.004X + 3	*p* = 0.91
ADP (*n* = 63)	0.96 (0.93–0.97)	Y = 1.4 (0.86 to 1.93) + 0.99 (0.9 to 1.08)X	0.89 (0.83–0.93)	−1.37 ± 1.5	(−4.3 to 1.5)	Y = −0.05X − 1	*p* = 0.26
D_2_O (*n* = 63)	0.94 (0.90–0.96)	Y = 0.91 (0.25 to 1.56) + 0.85 (0.76 to 0.95)X	0.94 (0.90–0.96)	0.15 ± 1.6	(−3.0 to 3.3)	Y = 0.06X − 0.2	*p* = 0.19
MRI (*n* = 7)	0.95 (0.69–0.99)	Y = −2.16 (−5.75 to–1.43) + 0.93 (0.57 to 1.28)X	0.76 (0.35–0.92)	2.79 ± 1.7	(−0.6 to 6.2)	Y = 0.02X + 3	*p* = 0.88
Male children							
SF (*n* = 54)	0.93 (0.89–0.96)	Y = 0.19 (−0.53 to 0.92) + 1.07 (0.9 to 1.19)X	0.91 (0.85–0.94)	−0.3 ± 1.6	(−3.5 to 2.9)	Y = −0.2X + 1	*p* < 0.001
DXA (*n* = 54)	0.95 (0.91–0.97)	Y = −2.22 (−3.02 to–1.43) + 0.96 (0.87 to 1.04)X	0.8 (0.71–0.86)	2.59 ± 1.4	(−0.1 to 5.3)	Y = −0.009X + 3	*p* = 0.84
ADP (*n* = 54)	0.96 (0.93–0.98)	Y = 0.61 (0.09 to 1.14) + 0.98 (0.9 to 1.06)X	0.95 (0.92–0.97)	−0.57 ± 1.2	(−2.9 to 1.8)	Y = −0.01X − 0.5	*p* = 0.74
D_2_O (*n* = 54)	0.96 (0.93–0.98)	Y = 0.77 (0.23 to 1.31) + 0.91 (0.83 to 0.99) X	0.96 (0.93–0.97)	−0.17 ± 1.2	(−2.6 to 2.3)	Y = 0.03X − 0.3	*p* = 0.53
MRI (*n* = 6)	0.94 (0.77–0.99)	Y = −2.14 (−4.41to 0.13) + 1.05 (0.73 to 1.37)X	0.92 (0.56–0.99)	−1.82 ± 1.5	(−1.2 to 4.8)	Y = −0.1X + 3	*p* = 0.54
Female adolescents							
SF (*n* = 92)	0.93 (0.90–0.95)	Y = 1.98 (0.59 to 3.4) + 0.92 (0.83 to 1.01)X	0.91 (0.88–0.94)	−1.2 ± 2.7	(−6.4 to 4.0)	Y = −0.09X + 0.01	*p* = 0.03
DXA (*n* = 92)	0.96 (0.94–0.97)	Y = −3.79 (−4.97 to−2.61) + 1.04 (0.98 to 1.1)X	0.87 (0.83–0.91)	3.13 ± 2.0	(−0.9 to 7.1)	Y = −0.08X + 4	*p* = 0.01
ADP (*n* = 92)	0.96 (0.94–0.98)	Y = 1.89 (1.05 to 2.73) + 1.02 (0.96 to 1.08)X	0.92 (0.89–0.95)	−2.10 ± 2.0	(−6.0 to 1.8)	Y = −0.06X − 1	*p* = 0.06
D_2_O (*n* = 92)	0.95 (0.93–0.97)	Y = 1.38 (0.38 to 2.37) + 0.86 (0.8 to 0.92)X	0.94 (0.91–0.96)	0.79 ± 2.6	(−4.3 to 5.8)	Y = 0.1X − 0.9	*p* < 0.001
MRI (*n* = 17)	0.86 (0.65–0.95)	Y = −1.83 (−7.5 to 3.83) + 1.03 (0.7 to 1.36)X	0.75 (0.53–0.88)	1.37 ± 2.9	(−4.3 to 7.0)	Y = −0.2X + 4	*p* = 0.2
Male adolescents							
SF (*n* = 79)	0.88 (0.82–0.92)	Y = 1.16 (−0.001 to 2.32) + 0.86 (0.76 to 0.96)X	0.84 (0.77–0.90)	0.74 ± 2.9	(−4.8 to 6.3)	Y = −0.1X + 2	*p* = 0.09
DXA (*n* = 79)	0.89 (0.83–0.93)	Y = −1.56 (−3.01 to 0.12) + 0.87 (0.77 to 0.97)X	0.77 (0.68–0.83)	3.16 ± 2.9	(−2.5 to 8.8)	Y = 0.02X + 3	*p* = 0.68
ADP (*n* = 79)	0.92 (0.88–0.95)	Y = 1.89 (0.97 to 2.8) + 0.89 (0.8 to 0.98)X	0.87 (0.81–0.92)	−1.00 ± 2.5	(−5.9 to 3.9)	Y = 0.04X − 1	*p* = 0.49
D_2_O (*n* = 79)	0.86 (0.78–0.91)	Y = 1.13 (−0.09 to 2.35) + 0.8 (0.7 to 0.9)X	0.84 (0.76–0.89)	0.98 ± 3.3	(−5.6 to 7.5)	Y = 0.08X + 0.2	*p* = 0.24
MRI (*n* = 19)	0.86 (0.65–0.95)	Y = −3.3 (−7.98 to 1.38) + 1.02 (0.67 to 1.37)X	0.6 (0.28–0.80)	3.04 ± 3.2	(−3.3 to 9.4)	Y = −0.X + 6	*p* = 0.14

SF = skinfold thickness, DXA = dual X-ray absorptiometry, ADP = air-displacement plethysmography, D_2_O = deuterium oxide dilution, MIR = magnetic resonance imaging, 4-C model = four compartments model, CI = confidence interval, SD = standard deviation, BA = Bland–Altman LOA = limits of agreement.

**Table 3 nutrients-14-01073-t003:** Correlation, concordance, agreement, and proportional bias assessment between SF, DXA, ADP, D_2_O and MRI with respect to 4-C model for the estimation of fat free mass (FFM).

Method	Pearson’s R, (95% CI)	Regression Equation	Lin’s Concordance Coefficient (95% CI)	Bland–Altman Difference Mean ± SD	BA LOA	Regression Analysis from Bland–Altman Differences	*p* Value
Total sample							
SF	0.98 (0.97–0.98)	Y = 0.95X + 2.1	0.98 (0.97–0.98)	0.31 ± 2.4	(−4.5 to 5.1)	Y = −0.03X + 1	*p* = 0.006
DXA	0.98 (0.98–0.99)	Y = 0.94X − 1.1	0.95 (0.94–0.96)	−2.97 ± 2.2	(−7.2 to 1.3)	Y = −0.04X − 2	*p* < 0.001
ADP	0.98 (0.98–0.99)	Y = 0.98X + 1.8	0.98 (0.97–0.98)	1.28 ± 2.2	(−3.1 to 5.7)	Y = −0.0005X + 1	*p* = 0.96
D_2_O	0.98 (0.97–0.98)	Y = 0.96X + 0.83	0.98 (0.97–0.98)	−0.53 ± 2.5	(−5.4 to 4.3)	Y = −0.02X + 0.2	*p* = 0.08
MRI	0.97 (0.94–0.99)	Y = 0.92X + 0.80	0.97 (0.95–0.98)	−2.28 ± 2.8	(−7.2 to 3.2)	Y = −0.04X − 0.9	*p* = 0.25
Female children							
SF	0.95 (0.93–0.97)	Y = 1.1X − 0.93	0.94 (0.91–0.96)	0.46 ± 1.6	(−2.7 to 3.6)	Y = 0.1X − 2	*p* = 0.004
DXA	0.96 (0.93–0.97)	Y = 0.90X − 0.77	0.8 (0.72–0.86)	−2.84 ± 1.3	(−5.5 to −0.2)	Y = −0.07X − 2	*p* = 0.08
ADP	0.94 (0.91–0.97)	Y = 1.0X + 0.35	0.91 (0.86–0.94)	1.28 ± 2.2	(−3.0 to 5.6)	Y = 0.004X + 1	*p* = 0.74
D_2_O	0.95 (0.92–0.97)	Y = 1.0X − 0.62	0.95 (0.91–0.97)	−0.12 ± 1.6	(−3.3 to 3.0)	Y = 0.08X − 2	*p* = 0.06
MRI	0.91 (0.48–0.99)	Y = 0.91X − 0.95	0.81 (0.36–0.95)	−2.79 ± 1.7	(−6.2 to 0.63)	Y = 0.009X − 3	*p* = 0.96
Male children							
SF	0.95 (0.92–0.97)	Y = 1.0X − 0.54	0.95 (0.91–0.97)	0.22 ± 1.6	(−2.9 to 3.3)	Y = 0.09X − 2	*p* = 0.03
DXA	0.96 (0.94–0.98)	Y = 0.86X + 0.61	0.82 (0.75–0.88)	−2.7 ± 1.3	(−5.2 to −0.04)	Y = −0.1X − 0.3	*p* = 0.004
ADP	0.96 (0.93–0.98)	Y = 0.94X + 1.7	0.95 (0.92–0.97)	0.42 ± 1.36	(−2.2 to 3.1)	Y = −0.007X + 0.6	*p* = 0.85
D_2_O	0.97 (0.94–0.98)	Y = 0.97X + 1.0	0.97 (0.94–0.98)	0.21 ± 1.2	(−2.2 to 2.5)	Y = 0.005X + 0.08	*p* = 0.87
MRI	0.96 (0.66–0.99)	Y = 1.0X − 2.0	0.73 (0.26–0.92)	−2.82 ± 1.5	(−4.8 to 1.2)	Y = 0.05X − 3.0	*p* = 0.73
Female adolescents							
SF	0.90 (0.85–0.93)	Y = 0.89X + 5.1	0.89 (0.83–0.92)	1.14 ± 2.6	(−4.0 to 6.3)	Y = −0.01X + 1	*p* = 0.79
DXA	0.95 (0.92–0.97)	Y = 0.87X + 1.6	0.83 (0.77–0.87)	−3.27 ± 2.0	(−7.2 to 0.7)	Y = −0.09X − 0.02	*p* = 0.009
ADP	0.94 (0.91–0.96)	Y = 0.91X + 5.5	0.89 (0.85–0.93)	1.28 ± 2.3	(−3.2 to 5.8)	Y = −0.001X + 1	*p* = 0.005
D_2_O	0.93 (0.89–0.95)	Y = 0.98X − 0.29	0.92 (0.88–0.95)	−0.86 ± 2.5	(−5.8 to 4.0)	Y = 0.06X − 3	*p* = 0.14
MRI	0.88 (0.70–0.96)	Y = 1.0X − 1.4	0.86 (0.66–0.94)	2.79 ± 1.7	(−0.6 to 6.2)	Y = 0.02X + 3	*p* = 0.88
Male adolescents							
SF	0.96 (0.94–0.97)	Y = 0.84X + 6.1	0.95 (0.92–0.97)	−0.74 ± 2.8	(−6.2 to 4.7)	Y = −0.1X + 5	*p* < 0.001
DXA	0.95 (0.93–0.97)	Y = 0.85X + 3.7	0.90 (0.85–0.93)	−3.0 ± 3.1	(−9.0 to 3.0)	Y = −0.1X + 2	*p* = 0.001
ADP	0.95 (0.93–0.97)	Y = 0.88X + 6.2	0.94 (0.90–0.96)	1.03 ± 2.7	(−4.3 to 6.3)	Y = −0.07X + 4	*p* = 0.04
D_2_O	0.94 (0.91–0.96)	Y = 0.93X + 2.3	0.95 (0.92–0.97)	−0.80 ± 3.3	(−7.3 to 5.7)	Y = −0.009X − 0.4	*p* = 0.82
MRI	0.96 (0.90–0.99)	Y = 0.84X + 3.9	0.89 (0.65–0.97)	−3.04 ± 3.2	(−9.4 to 3.3)	Y = −0.1X + 3	*p* = 0.06

SF = skinfold thickness, DXA = dual X-ray absorptiometry, ADP = air-displacement plethysmography, D_2_O = deuterium oxide dilution, MRI = magnetic resonance imaging, 4-C model = four compartments model, CI = confidence interval, SD = standard deviation, LOA = limits of agreement.

## Data Availability

Not applicable.

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
