# Peer review of "Body Composition Assessment in Mexican Children and Adolescents. Part 1: Comparisons between Skinfold-Thickness, Dual X-ray Absorptiometry, Air-Displacement Plethysmography, Deuterium Oxide Dilution, and Magnetic Resonance Imaging with the 4-C Model"

_nutrients, 2022, doi:10.3390/nu14051073_

Round 1
Reviewer 1 Report
This is a cross sectional study comparing body composition results from five different methods (DXA, ADP, Deuterium Dilution, MRI, and skinfolds) with the assumed gold standard 4-C model. The techniques are considered to be improvements over BMI for obesity assessment. Significant differences from the 4-C model are noted. No technique is recommended, and the authors conclude that the methods are not interchangeable – not a novel finding.
Some specific comments and questions:
TITLE
The “Part 1” in the title, while perhaps intriguing, calls for some explanation of future directions. This does not seem to be addressed in the article, thus there seems to be no reason to include this phrase in the title.
ABSTRACT
Regarding “The clinical context in which BC is evaluated usually defines the ideal model to be used”: This is somewhat confusing. Perhaps a simpler definition or specification could be used here, as clinical ‘context’ more often dictates the availability of methods, not necessarily models. Cost, manpower, location, all play a role in driving the selection of BC techniques.
INTRODUCTION
P2 L61: FFM is obtained via the 4C model – it is not a surrogate. Also, the difference(s) between FFM and LM definitions are nuanced and complex and should either be defined, or alternate text should be used here. It seems that the authors are simply trying to point out that use of the 4C model is not common and that other methods are normally used. Best to clarify this statement.
P2 L69: The aim stated here does not follow the logic of the preceding statements in the paragraph, i.e., what does comparing BC methods have to do with lack of healthcare or technology? Is the focus then on finding low cost, easy to apply methods? The aim (and intent) of the study seems too simplistic as stated here – merely comparing methods. How would the results be applied to the healthcare situation in Mexico? These topics should be tied together or an alternative motivation needs to be presented.
MATERIALS AND METHODS
P3 L94: This is awkwardly worded. Who was asked to assent?
P3 L125: DXA: Was the head ROI excluded from the analyses (as per recommendation by the ISCD)? If so, it should be stated. If not, DXA analysis should be redone and all relationships recalculated.
P5 L176: “(n=52??)” Is there uncertainty of the number of participants or is this merely a typo?
P5 L191: TBW should be in liters, not kg.
P5 L207: The means of FM would be compared using t-test, not calculated.
RESULTS
Table 1: The n values in the column headers add up to 293 rather than 288. Are the n values incorrect or is the entire cohort presented here?
Table 1; Body Composition Variables: The mean values do not all agree with those in supplementary Table 1.
Table 1; MRI subsample: There is no indication of age, BMI, etc. in this group. That information may be helpful, perhaps also as supplementary data.
P7 L226: It would be helpful if it was made clear that this refers to comparisons of means of all subjects (not broken down by age, sex).
P12 L283: Is Table 4 necessary? The title (and aim?) of the paper is all about comparing techniques against the 4C model. One also begins to wonder about affecting type 1 errors due to multiplicity of comparisons (debatable but worth considering).
It may be worth considering moving Table 4 to the supplementary file and bringing supplementary Table 1 into the main document. If the focus (see the title) of the article is comparisons with the 4C model, then supplementary Table 1 seems to merit more direct attention.
P13 L294: The legend for Figure 3 seems incorrect. It matches that for Figure 1.
As stated above, it may be worth considering moving these Bland-Altman figures to the supplementary file.
DISCUSSION
P15 L393: This paragraph reads more like a conclusion and seem out of place. It should perhaps be the last paragraph in the Discussion.
OVERALL
Much of the Discussion addresses strengths and weaknesses of the different BC methods, as if to provide guidance for clinicians in selecting the best techniques according to their needs and capabilities. None of the methods are ideal – all are flawed (in comparison with a gold standard). Yet a firm conclusion seems to be lacking. Concluding that methods differ is not surprising. One may be better served by discussing how the differences in methods (i.e. over or under estimating FM) may affect health assessments. That is, what impact may underestimating FM, by relying on one particular method, have on child care? How critical is the method selection? Somehow this needs to be tied more firmly to the population and environment being studied.
Author Response
We, the authors, want to express our gratitude to the Editorial Board and the experts that reviewed our manuscript for their contributions. We have found your recommendations and commentaries of great value and have made appropriate changes. We feel confident that the manuscript has been strengthened by this.
Please find our answers in the following lines.
TITLE
- The “Part 1” in the title, while perhaps intriguing, calls for some explanation of future directions. This does not seem to be addressed in the article, thus there seems to be no reason to include this phrase in the title.
The second part of this analysis, titled: “Body composition assessment in Mexican children and adolescents. Part 2: Cross-validation of three bio-electrical impedance methods against dual X-ray absorptiometry for whole-body and regional body composition”, has already been accepted by Nutrients #1604578. The two analyses were conducted on the same children’s database and closely complement each other. As the accepted manuscript already includes the phrase ‘Part 2’ we prefer to keep ‘Part 1’ here
ABSTRACT
2. Regarding “The clinical context in which BC is evaluated usually defines the ideal model to be used”: This is somewhat confusing. Perhaps a simpler definition or specification could be used here, as clinical ‘context’ more often dictates the availability of methods, not necessarily models. Cost, manpower, location, all play a role in driving the selection of BC techniques.
Changed, line 19
INTRODUCTION
3. P2 L61: FFM is obtained via the 4C model – it is not a surrogate. Also, the difference(s) between FFM and LM definitions are nuanced and complex and should either be defined, or alternate text should be used here. It seems that the authors are simply trying to point out that use of the 4C model is not common and that other methods are normally used. Best to clarify this statement.
Corrected, we adjusted this paragraph together with the next one to give a clearer explanation of the motivation of this study.
4. P2 L69: The aim stated here does not follow the logic of the preceding statements in the paragraph, i.e., what does comparing BC methods have to do with lack of healthcare or technology? Is the focus then on finding low cost, easy to apply methods? The aim (and intent) of the study seems too simplistic as stated here – merely comparing methods. How would the results be applied to the healthcare situation in Mexico? These topics should be tied together, or an alternative motivation needs to be presented.
We have adjusted accordingly. We hope now we can give a clearer explanation about the motivation of this study.
MATERIALS AND METHODS
5. P3 L94: This is awkwardly worded. Who was asked to assent?
Clarified in lines 164.
6. P3 L125: DXA: Was the head ROI excluded from the analyses (as per recommendation by the ISCD)? If so, it should be stated. If not, DXA analysis should be redone and all relationships recalculated.
The data used was total body with head, as recommended for the ISCD when using DXA for body composition instead of for bone densitometry. We clarified and referenced this in the text, lines 204-208.
7. P5 L176: “(n=52??)” Is there uncertainty of the number of participants or is this merely a typo?
This was a typo, we corrected it. Line 260.
8. P5 L191: TBW should be in liters, not kg.
Corrected.
9. P5 L207: The means of FM would be compared using t-test, not calculated.
We corrected (lines 287-288).
RESULTS
10. Table 1: The n values in the column headers add up to 293 rather than 288. Are the n values incorrect or is the entire cohort presented here?
Indeed, it was like that. We have changed it to show only the data of the 288 subjects (without the 5 outliers).
11. Table 1; Body Composition Variables: The mean values do not all agree with those in supplementary Table 1.
This was because of the inclusion of the 5 outliers in table 1, which has been corrected.
12. Table 1; MRI subsample: There is no indication of age, BMI, etc. in this group. That information may be helpful, perhaps also as supplementary data.
We have added this in supplementary table as S table 1.
13. P7 L226: It would be helpful if it was made clear that this refers to comparisons of means of all subjects (not broken down by age, sex).
Added, the analysis was for the total sample and by subgroups by age and sex, line 287
14. P12 L283: Is Table 4 necessary? The title (and aim?) of the paper is all about comparing techniques against the 4C model. One also begins to wonder about affecting type 1 errors due to multiplicity of comparisons (debatable but worth considering). It may be worth considering moving Table 4 to the supplementary file and bringing supplementary Table 1 into the main document. If the focus (see the title) of the article is comparisons with the 4C model, then supplementary Table 1 seems to merit more direct attention.
We have corrected as recommended. Table 4 sent to supplementary material as supplementary table 3.
15. P13 L294: The legend for Figure 3 seems incorrect. It matches that for Figure 1.
Corrected.
16. As stated above, it may be worth considering moving these Bland-Altman figures to the supplementary file.
Changed as supplementary figure 3.
DISCUSSION
17. P15 L393: This paragraph reads more like a conclusion and seem out of place. It should perhaps be the last paragraph in the Discussion.
Changed
OVERALL
Much of the Discussion addresses strengths and weaknesses of the different BC methods, as if to provide guidance for clinicians in selecting the best techniques according to their needs and capabilities. None of the methods are ideal – all are flawed (in comparison with a gold standard). Yet a firm conclusion seems to be lacking. Concluding that methods differ is not surprising. One may be better served by discussing how the differences in methods (i.e. over or under estimating FM) may affect health assessments. That is, what impact may underestimating FM, by relying on one particular method, have on child care? How critical is the method selection? Somehow this needs to be tied more firmly to the population and environment being studied.
We have stated in the conclusion that while individual methods show bias relative to the 4-component reference, the high correlations indicate that all the methods perform well in ranking individual children as having high or low FFM and fat mass. This ranking is itself very valuable in routine clinical care, particularly for longitudinal assessment. We have provided new evidence for the Mexican population that all the methods have utility for this purpose. Furthermore, some of the biases for individual methods can be resolved by the publication of method-specific reference data, whereby all data can be converted to method-specific z-scores. Publishing such reference data is a further aim of our project.
I hope that you find this work suitable for your Journal and I look forward to hearing from you.
Sincerely,
Dra. Desiree Lopez-Gonzalez

Reviewer 2 Report
My expertise on BC measurement methods and some of the included statistical methods is limited. However, the methods seem sound, and it seems that this study provides valuable evidence for those seeking to determine fat mass and fat free mass of children for clinical or research purposes. The discussion paragraphs laying out the pros and cons of each method were particularly helpful. My comments may reflect my ignorance, but I offer them nonetheless as other readers may also be confused by some sections as I was.
- I am not familiar with the Bland-Altman method, so was unclear when I read the methods section whether the text in lines 204-207 was explaining the Bland-Altman method or describing a separate procedure. I take from the results that those sentences were describing the Bland-Altman procedure, and if so, adding some text along the lines of "In this procedure..." would be helpful. As it was, I couldn't understand the description provided, and it didn't seem to match the figures, which were simply labeled as FM, so seemed to be a simple plots of FM using 2 methods on first read.
- Relatedly, it would have been helpful to have more explanatory titles for the plots--eg "Differential correlation between methods across levels of FM". And, the meaning of the trend line was very counter-intuitive, so an interpretive note under the plots would have been helpful. (E.g., "A positive trend indicates increasing underestimation of FM at high FM levels; a negative trend indicates increasing overestimation of FM at high FM levels.
- There is no description of the race and ethnic composition of the sample. I am not well-versed in the ethnic composition of Mexico, however, would it be helpful to know the degree of representation of, for example, youth who are Black, of Indigenous ancestry, and of European ancestry? Or are there other ethnic or cultural groups that should be represented?
- I didn't understand why the data differed between Table 1 and Supplementary Table 1. Why did Ns (and means/SDs) differ?
- Not all abbreviations are listed for the table footnotes (FFM, D2O, DXA, LM, BV, ADP)
Author Response
Reviewer two comments:
- I am not familiar with the Bland-Altman method, so was unclear when I read the methods section whether the text in lines 204-207 was explaining the Bland-Altman method or describing a separate procedure. I take from the results that those sentences were describing the Bland-Altman procedure, and if so, adding some text along the lines of "In this procedure..." would be helpful. As it was, I couldn't understand the description provided, and it didn't seem to match the figures, which were simply labeled as FM, so seemed to be a simple plots of FM using 2 methods on first read.
Added
2. Relatedly, it would have been helpful to have more explanatory titles for the plots--eg "Differential correlation between methods across levels of FM". And, the meaning of the trend line was very counter-intuitive, so an interpretive note under the plots would have been helpful. (E.g., "A positive trend indicates increasing underestimation of FM at high FM levels; a negative trend indicates increasing overestimation of FM at high FM levels.
Added to supplementary figure 1 and main figures 1 and 2
3. There is no description of the race and ethnic composition of the sample. I am not well-versed in the ethnic composition of Mexico, however, would it be helpful to know the degree of representation of, for example, youth who are Black, of Indigenous ancestry, and of European ancestry? Or are there other ethnic or cultural groups that should be represented?
Specified in limitations lines 993-997. Important to say is that ~89% of Mexican population is Hispanic. Only 6.6% belong to indigenous population and 5.9% to Afro-Mexican ethnic groups.
4. I didn't understand why the data differed between Table 1 and Supplementary Table 1. Why did Ns (and means/SDs) differ?
Corrected. In the previous version we put data of the whole sample of 293 subjects (including 5 outliers that should not been there). We have corrected the data, and now we only present data on the 288 subjects for both tables.
5. Not all abbreviations are listed for the table footnotes (FFM, D2O, DXA, LM, BV, ADP)
Added to table 1.
I hope that you find this work suitable for your Journal and I look forward to hearing from you.
Sincerely,
Dra. Desiree Lopez-Gonzalez
